# CD200 Induces Epithelial-to-Mesenchymal Transition in Head and Neck Squamous Cell Carcinoma via β-Catenin-Mediated Nuclear Translocation

**DOI:** 10.3390/cancers11101583

**Published:** 2019-10-17

**Authors:** Seung-Phil Shin, Ah Ra Goh, Hyeon-Gu Kang, Seok-Jun Kim, Jong-Kwang Kim, Kyung-Tae Kim, John H Lee, Yong-Soo Bae, Yuh-Seog Jung, Sang-Jin Lee

**Affiliations:** 1Division of Tumor Immunology, Research Institute & Hospital, National Cancer Center, Goyang 10408, Korea; apollossp@ncc.re.kr (S.-P.S.); ahra8714@naver.com (A.R.G.); 2Department of Biological Sciences, SRC Center for Immune Research on Non-lymphoid Organs, Sungkyunkwan University, Jangan-gu, Suwon 16419, Korea; 3Department of Biomedical Science, BK21-Plus Research Team for Bioactive Control Technology, College of Natural Sciences, Chosun University, 309 Pilmun-daero, Dong-gu, Gwangju 61452, Korea; kang84562@nate.com (H.-G.K.); heaven1472@chosun.ac.kr (S.-J.K.); 4Genome Analysis Team, Research Core Center, Research Institute & Hospital, National Cancer Center, Goyang 10408, Korea; jk@ncc.re.kr; 5Division of Cancer Biology, Research Institute & Hospital, National Cancer Center, Goyang 10408, Korea; bioktkim@ncc.re.kr; 6Adult Medical Affairs, NantKwest, 9020 Jefferson Blvd, Culver City, CA 90232, USA; john.lee@nantkwest.com; 7Center for Thyroid Cancer, Research Institute & Hospital, National Cancer Center, Goyang 10408, Korea

**Keywords:** HNSCC, CD200, EMT, β-catenin

## Abstract

The membrane glycoprotein CD200 binds to its receptor CD200R1 and induces tolerance, mainly in cells of the myeloid lineage; however, information regarding its role in solid tumors is limited. Here, we investigated whether CD200 expression, which is enriched mainly in high-grade head and neck squamous cell carcinoma (HNSCC), correlates with cancer progression, particularly the epithelial-to-mesenchymal transition (EMT). The forced overexpression of CD200 in the HNSCC cell line, UMSCC84, not only increased the expression of EMT-related genes, but also enhanced invasiveness. The cleaved cytoplasmic domain of CD200 interacted with β-catenin in the cytosol, was translocated to the nucleus, and eventually enhanced EMT-related gene expression. CD200 increased the invasiveness of mouse tonsillar epithelium immortalized with E6, E7, and Ras (MEER), a model of tonsillar squamous cell carcinoma. siRNA inhibition of CD200 or extracellular domain of CD200R1 down-regulated the expression of EMT-related genes and decreased invasiveness. Consistently, compared to CD200-null MEER tumors, subcutaneous CD200-expressing MEER tumors showed significantly increased metastatic migration into draining lymph nodes. Our study demonstrates a novel and unique role of CD200 in inducing EMT, suggesting the potential therapeutic target for blocking solid cancer progression.

## 1. Introduction

Head and neck squamous cell carcinoma (HNSCC) is a major cause of cancer-related deaths worldwide [1]. Although tobacco and alcohol consumption are the traditional risk factors for HNSCC, infection by human papillomavirus (HPV) has recently emerged as another major risk factor, especially for carcinomas of the oropharynx in developed countries [2,3,4]. Furthermore, the survival rate decreases with the progression of HNSCC to severe phenotypes [5]. Tumor differentiation is usually associated with cancer biology in HNSCC, and generally, high-grade tumors exhibit more progressive phenotypes than their low-grade counterparts [6]. In agreement with this, ‘dedifferentiation’ has been associated with features such as cancer stemness, partial epithelial–mesenchymal transition (EMT) and EMT, which are related to metastasis and recurrence [7]. Yet, it is imperative that epithelial tumor spread and non-classical EMT genes expressed in the tumor microenvironment are further investigated for better understanding the p-EMT process in the HNSCC tumor microenvironment.

The membrane glycoprotein CD200 was originally identified as an immune tolerance-signaling molecule that mainly regulates myeloid lineages [8,9] and was eventually associated with cancer progression [10,11,12]. Upon interaction with its receptor CD200R1, CD200 triggers an immunosuppressive signal, leading to macrophage inhibition, regulatory T-cell induction, cytokine profile switching from Th1 to Th2, and eventually, inhibition of tumor-specific T-cell immunity [12,13]. In addition to this aforementioned ‘canonical’ action of CD200 as an immune checkpoint, a ‘non-canonical’ role for CD200 has been occasionally reported [14,15,16]. These reports have shown that CD200 is co-expressed with cancer stem cell markers in prostate (CD44), breast (CD44 + CD24-), and colon (CD133) cancer cells, as well as in glioblastoma (CD133) cells [17]; furthermore, CD200-expressing human basal cell carcinoma cells were shown to initiate tumor growth [18] and were resistant to etoposide [19]. In agreement with these observations, our group reported that CD200 overexpression rendered mouse tonsil cancer cells resistant to chemoradiation [15]. Although CD200 triggers the signaling inside tumor cells, the mechanisms contributing to treatment resistance and tumor initiation beyond its established immune-regulatory functions remain unclear. EMT, invasive ability, and cancer stemness, which are connected to dedifferentiation, are critical features contributing to cancer progression, treatment resistance, and poor survival [20,21]. We herein report an intrinsic ‘non-canonical’ intracellular signaling pathway driven by the membrane protein CD200, whereby it exclusively induces ‘pro-tumor’ phenotypes. Our observations indicate the potential of targeting CD200 for therapy of patients with advanced HNSCC, especially after undergoing dedifferentiation and EMT-related changes.

## 2. Results

### 2.1. CD200 Was Associated with EMT Features in HNSCC 

Previous studies have suggested that CD200 overexpression is associated with cancer stem cells and resistance to chemoradiation using HNSCC cell lines [15]. As the metastatic HNSCC exhibited significantly higher CD200 expression compared to the non-metastatic one, (Appendix A), here, we analyzed the genomic landscape associated with CD200 expression in the clinical dataset of HNSCC. The association between CD200 expression and clinical features of HNSCC patients with respect to advanced tumor grade was analyzed using TCGA. The results showed significant upregulation of the CD200 transcript in grade 4 HNSCC compared to in grade 1 NHSCC (*p* < 0.024) and in the normal group (*p* < 0.0358). Furthermore, patients with grade 2 and 3 tumors showed significantly higher CD200 expression than patients with grade 1 tumors (*p* < 0.005 and *p* < 0.0002, respectively) (Figure 1A). Overall, in the TCGA dataset, CD200 expression was closely associated with tumor grade, which is considered a marker of the extent of malignant growth. Next, we searched for HNSCC cell lines that endogenously overexpress CD200 to study CD200 function; however, CD200 expression in these cell lines was too low to be manipulated for functional studies. Hence, we established human HNSCC cells overexpressing CD200 using a recombinant lentivirus encoding a CD200 cassette. UMSCC47 harbors an HPV viral genome in the chromosome, but the rest of them do not. Indeed, compared to the all HNSCC/control cells, forced overexpression of CD200 in HNSCC/CD200^High^ dramatically induced invasiveness (Figure 1B–E). Compared to the HNSCC/control, HNSCC/CD200^High^ with CD200 overexpression showed downregulation of E-cadherin and upregulation of both N-cadherin and vimentin proteins regardless of the HPV status (Figure 1B–E). Furthermore, we tested whether CD200 was involved in EMT and invasiveness in NTERA-2 cells derived from a malignant embryonal carcinoma and endogenously overexpressing CD200. siRNA-mediated knockdown of *CD200* downregulated vimentin, weakly recovered E-cadherin expression (Appendix A), and dramatically reduced invasiveness (Appendix A). These observations were indicative of a CD200-triggered noncanonical cytoplasmic pathway, resulting in EMT and invasiveness, in addition to its previously well-known ‘canonical’ roles as a driver of immune cell tolerance. 

### 2.2. EMT after CD200 Overexpression in MEER Cells

As CD200 plays a critical role in immune cells, and as cancer cell metastasis involves the function of several immune cells, we further investigated the role(s) of CD200 during EMT of MEER cells established as a murine HPV+ tonsil carcinoma model [15]. MEER/CD200 cells were sorted using anti-CD200 antibodies into two populations: high CD200-expressing and low CD200-expressing cells, hereafter referred to as MEER/CD200^High^ and MEER/CD200^Low^, respectively (Figure 2A, left panel). To validate the noncanonical role(s) of CD200, such as acquisition of chemotherapy resistance and development of EMT, we assessed the invasiveness of MEER/CD200^Low^ and MEER/CD200^High^ cells on Transwell membranes coated with Matrigel. The MEER/CD200^High^ cells were more invasive than both MEER/CD200^Low^ and MEER/control cells (*p* = 0.039) (Figure 2A, right panel). As CD200 overexpression increases resistance to chemoradiotherapy and leads to the acquisition of cancer stem cell-like features [15], sphere formation and SOX2 expression, which are associated with stem cell pluripotency in HNSCC, were evaluated, [22,23]. The MEER/CD200^High^ cells exhibited more sphere formation under serum-free culture conditions than the MEER/CD200^Low^ and MEER/control cells (Appendix A). Simultaneously, MEER/CD200^High^ expressed more SOX2 transcripts and exhibited EMT features, as demonstrated by the enhanced invasive ability and resistance to cisplatin chemotherapy (Appendix A). Overall, our results suggest that CD200 is an EMT driver and endows cancer stem cell-like features to mouse OSCC cells. Quantification of the transcription (Figure 2B) and translation (Figure 2C) of EMT-related genes after CD200 overexpression in MEER cells revealed that MEER/CD200^High^ cells showed high levels of mesenchymal marker transcripts, such as vimentin, N-cadherin, whereas those of the epithelial marker E-cadherin were downregulated. We identified 186 EMT-related genes with >1.5-fold changes in transcription between MEER/control and MEER/CD200^High^ cells, which were then investigated in the TCGA HNSCC dataset. It is noteworthy that the EMT-related genes of MEER/CD200^High^ in CD200+ HNSCCs patients were regulated similarly to those in CD200- HNSCC patients (*p* < 0.05 for both fold change > 1.5 and < 0.5). This suggests a similar pattern of CD200-driven regulation of EMT in human and mouse HNSCCs (Figure 2D).

### 2.3. Changes in the EMT Features after CD200 Inhibition

To assess whether enhanced invasion and the elevated mesenchymal hallmarks were directly caused by CD200 overexpression, the MEER/CD200^High^ cells were treated with a CD200 siRNA. As shown in Figure 3A, the CD200 siRNA effectively downregulated CD200, which subsequently reduced the invasiveness of MEER/CD200^High^ cells (Figure 3B). A previous report showed that the cytoplasmic tail of CD200, released by γ-secretase cleavage, acts as a transcriptional regulator [24]. We assessed the invasiveness of the MEER cells in the presence of the γ-secretase inhibitor, DAPT. DAPT significantly reduced the invasiveness of MEER/CD200^High^ cells, whereas there was no effect on the invasiveness of the MEER/control (Figure 3C), confirming that CD200 cleavage was also necessary during EMT of MEER cells.

Next, we investigated whether CD200 can act as a target for blocking metastasis of HNSCC cells. An adenovirus harboring a gene expression cassette encoding the extracellular domain of CD200R1 under the EF1α promoter (Ad5sCD200R1) was constructed (Appendix A) and the resultant fusion protein contained the extracellular domain of CD200R1 (sCD200R1-Ig) (Appendix A). The MEER/CD200^High^ cells were infected with either Ad5sCD200R1 or Ad5MOCK; at 5 multiplicity of infection (MOI), Ad5sCD200R1 significantly reduced the invasiveness of MEER/CD200^High^ cells (Figure 3D). Similarly, when purified sCD200-Ig proteins produced from Ad5sCD200R1 infected-293 cells were added to MEER/CD200^High^ cells in the upper chamber of a Transwell, the soluble sCD200R1-Ig effectively inhibited invasiveness (Figure 3E). These results imply that the efficacy of CD200 cleavage by γ-secretase can be modulated by restricting the movement of the extracellular domain of CD200, which can act as a therapeutic target for blocking cancer cell metastasis.

### 2.4. Interaction of Cleaved CD200 with β-catenin in the Cytoplasm

To undercover the mechanism of CD200-driven EMT, we performed co-IP for identifying the proteins that bind to the cytoplasmic domain of CD200. We transfected a vector expressing the CD200 cytoplasmic domain fused with 3× FLAG tags at the C-terminal in UMSCC84 and observed that it was bound to β-catenin (Figure 4A). To determine whether the CD200 cytoplasmic-domain β-catenin complex translocated into the nucleus, nuclear and cytoplasmic extracts were separately prepared. Results confirmed the presence of the CD200 cytoplasmic tail and β-catenin in both fractions (Figure 4B) and the translocation of β-catenin into the nucleus upon CD200-overexpression in UMSCC84 cells (Figure 4C). Next, we investigated whether the β-catenin-CD200 cytoplasmic tail complex enhanced the transcriptional activity of β-catenin using the TCF/LEF-luciferase reporter system. As shown in Figure 4D, CD200 significantly increased luciferase activity. The expression of several target genes of β-catenin were also increased in UMSCC84/CD200^High^ cells (Figure 4E). Simultaneously, the treatment of CD200 overexpressing NTERA-2 cell with the CD200 siRNA decreased the expression of the β-catenin target genes (Appendix A). To further verify the effect of γ-secretase on CD200, we exposed UMSCC84/CD200^High^ to DAPT and observed a significant reduction in the invasiveness of UMSCC84/CD200^High^ cells and a reversion of the EMT-related gene expression pattern that was caused with CD200-overexpression (Figure 4F), suggesting that the CD200 tail domain cleaved by γ-secretase was bound to β-catenin, which mediates its translocation into the nucleus and eventually, increases the expression of EMT-related genes (Figure 4G).

### 2.5. CD200-Induced Increase in Metastatic Ability in an Animal Model

To further determine whether CD200 overexpression facilitates metastasis in vivo, we subcutaneously implanted MEER/control, MEER/CD200^Low^, and MEER/CD200^High^ cells on C57BL6 mouse flanks. Tumor growth in each group increased proportionately with CD200 expression and most tumors in the MEER/control group were rejected (tumor-free: 4/5 mice) (Figure 5A). After the tumors had grown to 200–300 mm^3^, the DLNs around the tumors were collected and analyzed immunohistochemically using the anti-CK18 antibody. Therefore, we histologically counted the number of metastatic cells in the DLNs containing MEER/CD200^High^ tumors, assuming that MEER/CD200^Low^ tumors have a histological score of 1. Mice bearing MEER/CD200^High^ tumors exhibited more metastatic cells in DLNs, confirming that CD200 significantly promotes metastases to DLNs (Figure 5B).

## 3. Discussion

As EMT, invasive ability, and cancer stemness are usually associated with dedifferentiation and are well-known phenotypes contributing to a poor therapeutic response to contemporary treatments, we attempted to determine the role of CD200 in the evolutional induction of chemoresistance in human and mouse HNSCC models. We showed that CD200 exclusively induced EMT, invasion, metastasis, and cancer stemness, which was in agreement with the association of CD200 with cancer stemness and the eventual development of in vivo resistance to chemoradiation [15]. Notably, CD200 was originally identified as a tolerogenic immune-checkpoint factor involved mainly in hematological malignancies. Considering that immune evasion mechanisms play a key role in HNSCC pathogenesis, we sought to expand our understanding of the intrinsic ‘noncanonical’ role of CD200 using human and mouse HNSCC models owing to the high homology (77.6%) in the amino acid sequence of human and mouse CD200 [25]. An immune-competent mouse model was more appropriate for evaluating the function of this specific oncogene as cancer progression, including metastasis and drug-resistance, often involves immune cells [26,27]. Although CD200 expression has been previously described in various human cancer cells, information regarding the association between CD200 expression and the progression of HNSCC is limited. The TCGA dataset revealed that CD200 expression was significantly high in patients with grade IV (*p* = 0.024) and grade III disease (*p* = 0.0002). Furthermore, we observed that CD200 overexpression induced resistance to cisplatin and sphere formation. These observations prompted us to investigate whether CD200 plays a unique and novel role by triggering EMT, stemness, invasion, and consequent cancer progression in both humans and mice. Our data consistently supported a novel role for CD200 in triggering this phenotypical transformation. First, CD200-overexpressing cells exhibited significantly enhanced invasiveness and a concordant shift toward a mesenchymal expression profile, including the upregulation of vimentin and N-cadherin and downregulation of the epithelial marker E-cadherin. Furthermore, the signal induction mechanism was unique as the mesenchymal transition was reversed by both siRNA and the soluble extracellular domain of CD200R1 (sCD200R1). Although the involvement of CD200 in EMT induction and invasiveness was evident, information regarding the underlying molecular mechanism driving this mesenchymal transition downstream of CD200 was scarce. To determine the molecular mechanism underlying EMT, we conducted co-IP and confirmed binding of the cytoplasmic tail of CD200 with β-catenin, followed by summarizing the research report of EMT and the whole RNA sequencing of MEER/CD200^High^. This observation supported the results of a previous study showing that the cytoplasmic tail of CD200 was cleaved by γ-secretase and translocated inside the nucleus [24]. As β-catenin is a key regulator of the mesenchymal transition in cancer [28,29], we investigated the relationship between CD200 and β-catenin and the resulting changes in EMT and invasiveness. Indeed, CD200 transduced a cellular signal that resulted in the expression of β-catenin target genes, which triggered EMT. Furthermore, the mode of action via which CD200 activates β-catenin was investigated, implying that the cytoplasmic domain of CD200 might release β-catenin molecules from the complex of CK1/GSK3/β-catenin for nuclear translocation. CD200 possesses a short cytoplasmic tail (human: nine amino acids, mouse: 19 amino acids), which is unlikely to recruit other signaling molecules to the cytoplasm. However, we reasoned that the short cytoplasmic tail of CD200 might recruit other molecules, similar to PD-L1 [30]. As blocking of CD200 with the CD200 siRNA or sCD200R1 dramatically and consistently reversed its ability to promote EMT, therapeutic targeting of CD200 might be effective for treating patients with solid tumors such as HNSCC. As our results indicate that CD200 induces EMT, invasion, and eventual resistance to chemotherapy, treatment strategies aimed at blocking CD200 might have the potential to overcome chemoresistance in particular. Although we used a strategy of blocking CD200 in the local microenvironment using sCD200R1, an anti-CD200 monoclonal antibody is traditionally used for targeting CD200 [31]. The efficacy of these different agents for solid cancers such as aggressive HNSCC should be tested, especially in the setting of combination therapy. Further studies are needed to determine whether the function of CD200 observed in HNSCC is applicable to various carcinomas.

CD200 possesses multiple properties, including a ‘protumor’ role, where it induces the ‘canonical’ immune checkpoints and the ‘noncanonical’ pathways leading to EMT and chemoresistance. Therefore, strategies targeting CD200 might be developed as new solid cancer therapeutics, especially in combination with contemporary treatments in HNSCC.

## 4. Materials and Methods

### 4.1. Cell Line and Culture

The HNSCC tumor cells were maintained in Dulbecco’s modified Eagle’s medium (DMEM) supplemented with 10% fetal bovine serum (FBS) and 1% penicillin/streptomycin. Each HNSCC cell in the study was prepared at the same culture passage and was used for all experiments. MEER cell lines were used as a murine model for OSCCs [32]. MEER/CD200 cells were established by stably transfecting cells with the pUNO1 mCD200 plasmid (Invitrogen, San Diego, CA, USA) [15]. MEER, MEER/CD200^High^, MEER/CD200^Low^, and NTERA-2 cells were cultured in DMEM containing 10% FBS.

### 4.2. Analysis of The Cancer Genome Atlas (TCGA) Data 

To investigate CD200 mRNA expression in HNSCCs, we acquired expression data and clinical information from the TCGA data portal on March 2018 and retrieved mRNA expression datasets from 44 normal tissues and 63 Grade 1, 305 Grade 2, 125 Grade 3, and 7 Grade 4 tumor-matched samples. The RNA expression data were normalized using RNA-seq expression estimations via expectation maximization (RSEM) [33] with a custom-made script. The normalized read counts from MEER/control and MEER/CD200^High^ mRNA expression data were obtained using an Illumina Nextseq 500 system (Illumina, San Diego, CA, USA). Fold changes were calculated to select genes that were differentially expressed as "high" vs. control (>1.5 or <0.5). The differentially expressed genes (DEGs) that overlapped with the TCGA mRNAseq data were selected as markers, and a heat map was generated using the z-scores of their normalized expression. TCGA mRNAseq data (provisional) from 522 head and neck carcinomas were downloaded from cBioPortal [33,34] and were normalized to the estimate value (level 3) from Illumina HiSeq. For each dataset, samples were dichotomized into up- and downregulated groups based on the median of CD200 expression. Student’s *t*-test and fold change were used to compare expression differences between the two groups. *p* value < 0.05 and fold change > 2 (or <0.5) were considered statistically significant. 

### 4.3. Western Blot Analysis and Flow Cytometry 

The cells were washed with phosphate-buffered saline (PBS) and lysed in radioimmunoprecipitation assay (RIPA) buffer containing protease inhibitors (Sigma-Aldrich, St. Lois, MO, USA). Western blot analyses were conducted as previously described [15]. Antibodies to the following proteins were used: ZEB1 and ZEB2 (Novusbio, Littleton, CO, USA), cyclin-D1 (Merck, Darmstadt, Germany), N-Cadherin (Merck), E-cadherin (Merck), vimentin (Merck), CD200 (R&D systems, Minneapolis, MN, USA), CD200R1 (R&D systems), c-Myc (Cell Signaling Technology, Danvers, MA, USA), β-catenin (Merck), Fibronectin (Merck), and β-actin (Santa Cruz, Dallas, TX, USA). For flow cytometry analysis, all cells (1 × 10^7^) or dissociated tumors were incubated for 15 min in the dark with anti-mouse CD16/CD32 antibody (BD Biosciences, San Diego, CA, USA). Cells (1 × 10^7^) were incubated for 30 min with anti-human CD200 PE-Cy7-conjugated antibody (BD Biosciences) or anti-mouse CD200 PE-conjugated antibody (BioLegend, San Diego, CA, USA). After washing again with FACS buffer, the MEER cell lines were resuspended in FACS buffer and analyzed. Densitometry readings/intensity ratio of each band was performed by using ImageJ software (ImageJ, NIH, Bethesda, MD, USA). Details information of western blot could be found in Appendix A.

### 4.4. Co-Immunoprecipitation (co-IP)

Cell lysate extracts were prepared in immunoprecipitation (IP) buffer and the debris was removed via centrifugation. The supernatants were pre-cleared by treatment with anti-FLAG M2 agarose beads in an agitator, followed by centrifugation at 2000 RPM for 4 min at 4 °C (×3). The supernatants were immunoprecipitated with M2 agarose beads, and the immunoprecipitates were washed in IP buffer and subjected to Western blotting with anti-β-catenin or anti-FLAG antibodies. Cytosolic and nuclear fractions were prepared according to the manufacturer’s protocol (Sigma-Aldrich).

### 4.5. Adenovirus Construction 

AdenoZAP^TM^ 1.2 kits truncating E1 and E3 (OD260, Boise, ID, USA) were used to generate Ad5sCD200R1, which harbored the extracellular domain of mouse CD200 receptor 1 fused with mouse Fc1 of IgG2A (mIgG2A), called sCD200R1-Ig, under the control of the EF1α promoter. To generate sCD200R1-Ig, the extracellular domain of mouse CD200R1 (OriGene, Rockville, MD, USA) was amplified using a polymerase chain reaction (PCR) with primers (Appendix A). The amplified product was inserted into the *EcoR*I/*Nco*I site of pFUSE-mIgG2A.Fc1, which contains the EF1α promoter, the Fc1 of mIgG2A, and the SV40 poly A sequence (Invivogen, San Diego, CA, USA), resulting in sCD200R1 fused with Fc1 of IgG2a. Then, EF1α.sCD200R1 was subcloned into the *Not*I/*EcoR*V site of a viral shuttle vector, pZAP1.1 (OD260) to generate pZAP1.1.EF1a.sCD200R1.Fc1. To construct Ad5sCD200R1, pZAP1.1.EF1a.sCD200R1.Fc1 was digested with *Dra*III/*Pac*I/*Cla*I, ligated with RightZAP1.2 (OD260), and transfected in HEK293 cells. The control virus Ad5MOCK was prepared and used as described previously [35].

### 4.6. PCR and Quantitative Reverse Transcription PCR (qRT-PCR)

Total RNA was extracted from MEER/control, MEER/CD200^High^, and MEER/CD200^Low^ cells using TRIzol (Invitrogen) according to the manufacturer’s protocols. cDNA synthesis was performed in a 10 μL solution using Qiagen Omniscript RT-kits (Qiagen Sciences, Gaithersburg, MD, USA). PCR was performed using primers listed in Appendix A under the following conditions: 10 min at 95 °C for preheating; 30 cycles of 30 s at 95 °C, 30 s at 60 °C, and 30 s at 72 °C; 10 min at 72 °C. Next, qRT-PCR was performed using a QGreen master mix kit (CellSafe, Suwon, Korea) and predesigned primer/probe pairs for mouse or human CD200, E-cadherin, N-cadherin, and vimentin. Expression was normalized to that of GAPDH using the LightCycler software (Roche, Indianapolis, IN, USA). All measurements were performed in triplicate.

### 4.7. Transient Transfection

Cells were seeded at 2 × 10^5^ cells/well in 100 mm^2^ dishes and transiently transfected with siRNA targeting CD200 or scrambled siRNA (sc siRNA) (Origene Technologies, Inc., Rockville, MD, USA). For each transfection, 20 pmoles siRNA in 500 µL serum-free Opti-MEM mixed with 7 µL RNAiMAX (Invitrogen) was used. 

### 4.8. Immunocytochemistry (ICC) and Confocal Microscopy

MEER cells were cultured for 48 h with Flag-E.V or Flag-CD200 transfection in an 8-well chamber slide (SPL Life Sciences co., Pochenon, South Korea) and fixed with 3.7% formaldehyde. The cells were blocked with 5% BSA and then incubated for 2 h with β-catenin antibodies (Santa Cruz). The chamber slide was washed and then incubated with the secondary antibodies conjugated with Alexa flour 647 (Invitrogen, MA, USA). Samples were washed and treated with Mounting Medium with DAPI. The chamber slides were covered with a cover glass and examined using a confocal microscope (Carl Zeiss LSM510, Oberkochen, Germany).

### 4.9. Invasion Assay

Matrigel stock (10.2 mg/mL) (BD Biosciences) was diluted in serum-free DMEM medium to 0.5 mg/mL, and 100 µL diluted Matrigel was added to each Transwell containing an 8 μm pore size polycarbonate filter (BD Bioscience). After 12 h, the Transwells were ready for invasion assays. In total, 1 × 10^5^ cells were resuspended in serum-free DMEM and placed in the upper chambers of the Transwells. The lower chambers of the Transwells were filled with 750 µL DMEM containing 10% FBS. The cells in the upper chambers were incubated for 24, 48, or 72 h and then stained with crystal violet. Cells that adhered to the underside of the filter were counted in three randomly selected fields using a light microscope.

### 4.10. Animal Experiment

Female 6-week-old C57B6 mice (OrientBio, Sungnam, Korea) were inoculated subcutaneously with 1 × 10^6^ cells. When the tumor volume was 200–300 mm^3^, the largest draining lymph nodes (DLNs) around the tumors and lungs were harvested and fixed in 4% paraformaldehyde. Immunohistochemical staining for CK18+ was performed using the automated instrument Discovery XT (Ventana medical systems, Tucson, AZ, USA), rabbit anti-CK18 antibody (Abcam, Cambridge, UK), and an UltraMap anti-Rb HRP kit (Multimer HRP-conjugated anti-rabbit IgG). All animal experiments were performed in specific pathogen-free facilities and per the conditions of the Guidelines for the Care and Use of Laboratory Animals of the Korea National Cancer Center and the Association for Assessment and Accreditation of Laboratory Animal Care (approval number: NCC16-310).

### 4.11. Statistical Analysis

Comparisons between two groups were made using two-tailed paired *t*-tests. Two-tailed *p* values < 0.05 were considered statistically significant. The STATA/SE version 10.1 software (StataCorp LP, College Station, TX, USA) was used for analyses.

## 5. Conclusions

Taken together, CD200 exclusively modulates EMT in malignant tumors such as HNSCC. We found that the overexpression of CD200 was increased and the expression of EMT-related gene was upregulated as the malignant traits were obtained using TCGA data of HNSCC. Based on these results, we verified that the short tail of CD200 binds to β-catenin in cytosol and translocates into the nucleus and is involved in EMT gene expression. These results indicate the importance of understanding the transfer process of HNSCC as a new role of CD200 that has, to date, not been clarified.

## Figures and Tables

**Figure 1 cancers-11-01583-f001:**
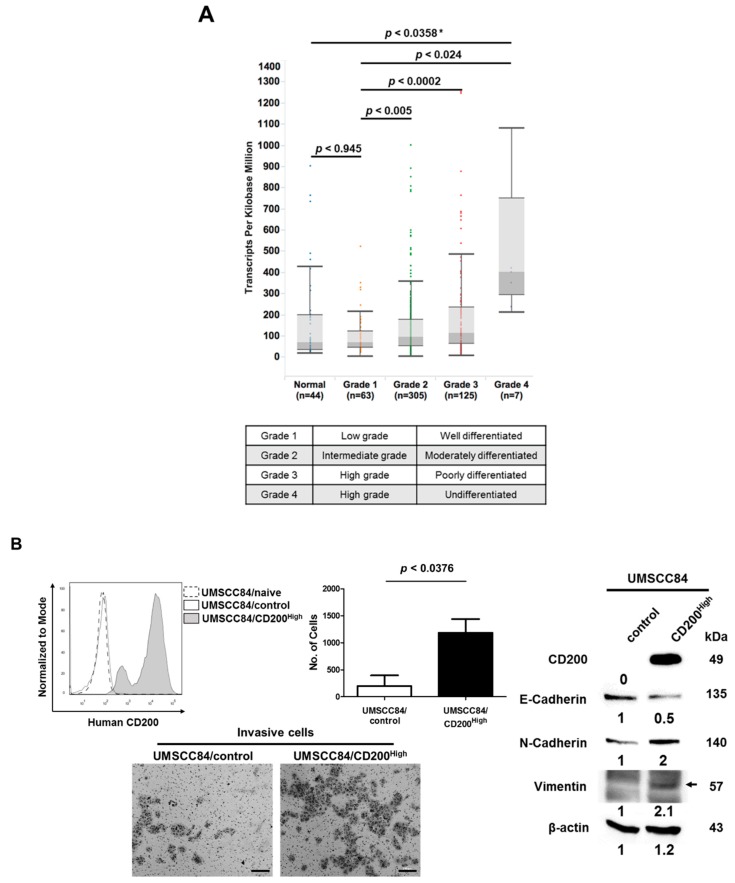
CD200 was upregulated in head and neck squamous cell carcinoma (HNSCC) patients and induced EMT. (**A**) CD200 mRNA expression in HNSCC patient samples in the TCGA dataset was analyzed by normalizing RNA-seq expression estimations by expectation maximization (RSEM). (**B**–**E**) UMSCC cell lines were transduced using a lentiviral vector carrying human *CD200* and were sorted into UMSCC/CD200^High^ and UMSCC/control cells (upper left). The UMSCC/control cell was used as the control. Cells (1 × 10^5^)/well of each cell line was placed on Matrigel-coated Transwells and incubated for 48 h. Cells migrating to the underside of the filter through the Transwell were stained with crystal violet (200 μm scale bar, lower panel) and counted under a microscope (upper right). Each error bar in the graph represents the average of three independent experiments (mean ± SEM). Protein (30 µg) was used for western blotting.

**Figure 2 cancers-11-01583-f002:**
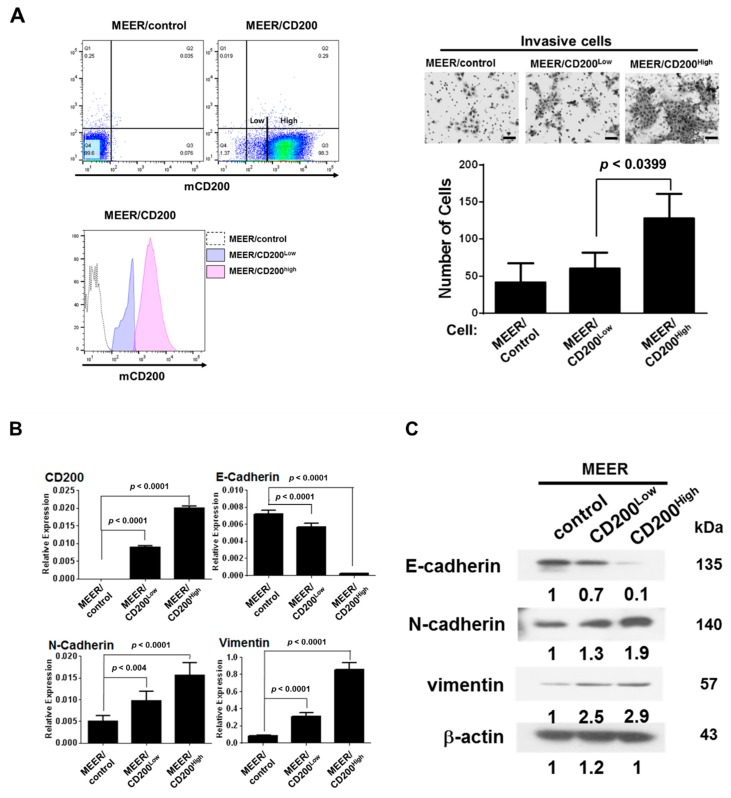
Mouse CD200 endowed MEER with better invasion ability by upregulating EMT-related genes, similarly to that observed in human CD200+ HNSCC patients. (**A**) MEER cells stably expressing CD200 were separated using flow cytometry into MEER/CD200^High^ and MEER/CD200^Low^ (left panel). MEER/control cells were established by transfecting cells with the empty vector pUNO1. Each cell type (1 × 10^4^ cells) were placed in the upper chamber of a Matrigel-coated Transwell for 24 h. Cells that migrated to the underside of the filter through the Transwell were stained with crystal violet and counted under a microscope (100 μm scale bar, right panel). Each error bar in the graph represents the average of three independent experiments (mean ± SEM). (**B**) Total mRNA from each cell type was assayed for the expression of EMT-related genes and their expression was normalized to that of mouse GAPDH RNA (mean ± SEM; *n* = 3). (**C**) Protein lysates (30 µg) were prepared from each cell type and were analyzed using Western blotting. (**D**) TCGA mRNA data showing CD200 overexpression in HNSCC (*n* = 522) patients, which were compared using the fold change values of total mRNA for MEER/control and MEER/CD200^High^ cells (Fold change > 1.5 or < 0.5). Student’s *t*-test and fold change were used to compare expression differences between the two groups. *p* value < 0.05 and fold change > 2 (or < 0.5) were considered statistically significant.

**Figure 3 cancers-11-01583-f003:**
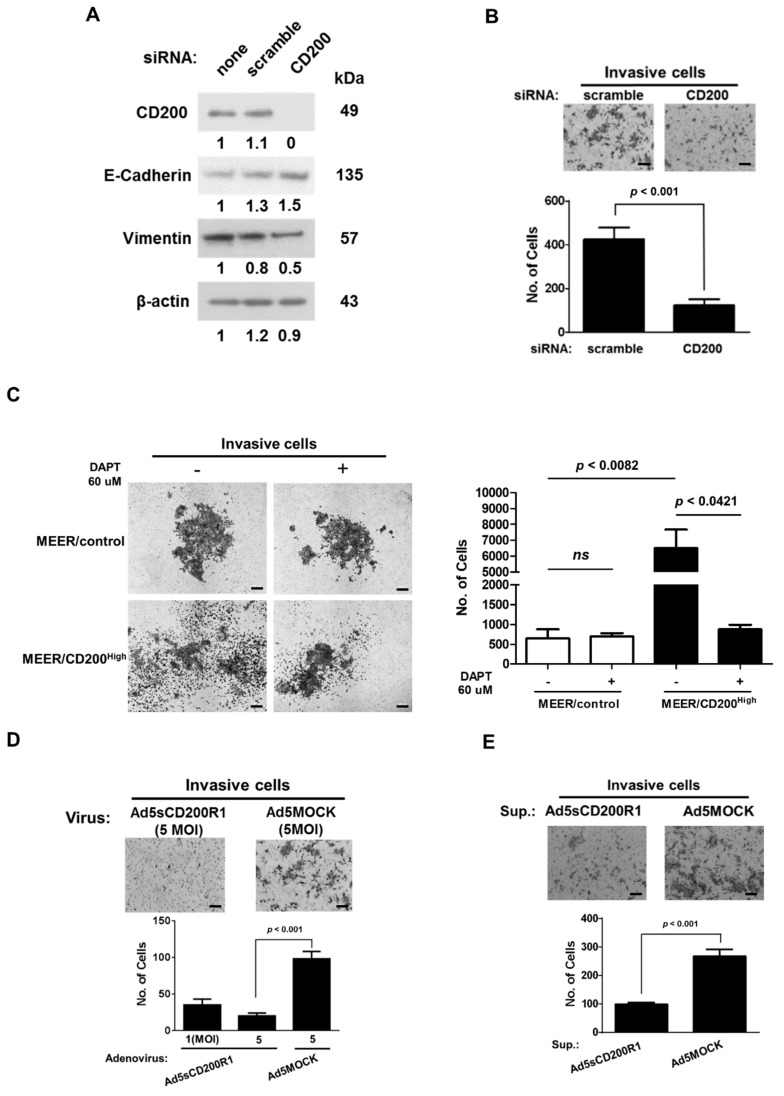
Inhibition of CD200 lowered EMT-related gene expression and reduced invasive ability. (**A**) MEER/CD200^High^ (2 × 10^5^) cells in a 100 mm^2^ dish were treated with 20 pmoles of CD200 siRNA for 24 h. The cells were then lysed with RIPA buffer, and total proteins were extracted for subsequent analyses. Protein extracts (30 µg) were prepared for Western blot analysis. Samples treated with scrambled siRNA were included as controls. (**B**) MEER/CD200^High^ cells were treated with either scrambled siRNA or CD200 siRNA for 24 h and then laid on the Matrigel of the Transwells for invasion assays. MEER/CD200^High^ cells in the Transwells were incubated for 24 h and then stained with crystal violet (100 μm scale bar, mean ± SEM; *n* = 3). (**C**) MEER cell lines (1 × 10^5^ cells) were seeded into the upper chamber of Matrigel-coated Transwells and treated with 60 µM DATP, a γ-secretase inhibitor. After 72 h of incubation, the invasive cells were stained with crystal violet and counted under a microscope (200 μm scale bar, mean ± SEM; *n* = 3). (**D**) MEER/CD200^High^ cells (4 × 10^3^ cells/well) were placed in the upper chamber of the Transwells and infected with different doses of the adenoviruses. At 48 h post-infection, cells that had invaded the underside of the filter of the Transwell were stained with crystal violet and counted under a microscope (100 μm scale bar, mean ± SEM; *n* = 3). (**E**) Quantities of 1 μg of purified sCD200R1-Ig from HEK293FT cells infected with 5 multiplicity of infection (MOI) Ad5sCD200R1 or Ad5MOCK were added to each Transwell containing MEER/CD200^High^ cells. Invasion assays were performed as described above (100 μm scale bar, mean ± SEM; *n* = 3).

**Figure 4 cancers-11-01583-f004:**
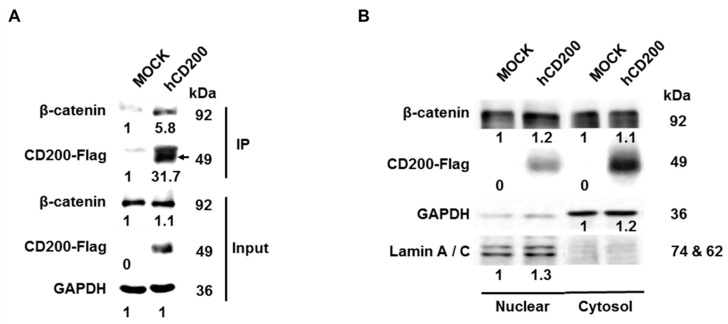
Cytoplasmic domain of CD200 interacted with β-catenin, induced the translocation of β-catenin in the nucleus, and increased EMT-related gene expression. (**A**) The interaction between the 3X-FLAG-tagged cytoplasmic domain of CD200 with β-catenin was investigated by transfecting the human CD200 overexpression vector into UMSCC84 cells, followed by immunoprecipitation. (**B**) CD200-FLAG and β-catenin levels were analyzed using Western blotting after preparing nuclear and cytosolic lysates from *CD200*-transfected UMSCC84. Lamin A/C and GAPDH were used as nuclear and cytosolic markers, respectively. (**C**) Detection in expression and localization of β-catenin in Flag-E.V or Flag-CD200 transfected into MEER cells. Immunocytochemistry were performed with anti-β-catenin Alexa flour 647 (red) followed by DAPI nuclear counterstainging (Blue) were detected by confocal microscopy. The bar indicates 10 μm. (**D**) TCF/LEF-luciferase activity was measured in UMSCC84 cells transfected with CD200 overexpression vector for 24 h. (**E**) UMSCC84 cells (1 × 10^5^) were cultured in 6-well plates for 48 h. The cell lysates (30 µg) were analyzed using Western blotting. (**F**) UMSCC84 cells (1 × 10^5^) were seeded in Matrigel-coated Transwell in the presence of 40 µM γ-secretase inhibitor for 48 h. Each error bar in the graph represents the average of three independent experiments (mean ± SEM). The cell lysates (30 µg) by γ-secretase inhibitor were analyzed using Western blotting. (**G**) The CD200 cytoplasmic domain is cleaved by γ-secretase within the transmembrane, which binds to β-catenin in the cytoplasm and is translocated into the nucleus. The anterior amino acids of the cytoplasmic tail of hVEGFR1, hNOTCH, and hCD200, which are cleaved by γ-secretase, have common positive charge (red color).

**Figure 5 cancers-11-01583-f005:**
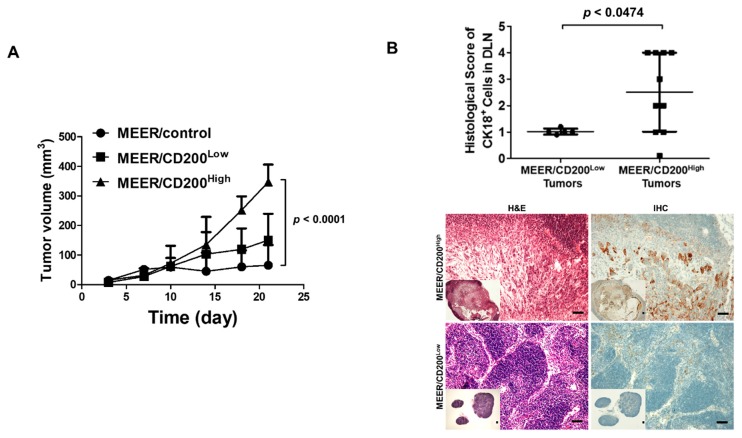
CD200 overexpression enabled MEER tumors to metastasize to DLNs. (**A**) One million MEER cells were subcutaneously implanted into C57BL6 mice. When the tumors were approximately of 200–300 mm^3^, the tumors, DLNs, and major organs were harvested. CD200 expression was analyzed to confirm expression differences between MEER/CD200^High^ and MEER/CD200^High^ tumors. The DLNs were sectioned to detect metastasis using an anti-CK18 antibody, and the numbers of metastases were histologically evaluated. (**B**) The frequency of metastasis in mouse DLNs was 100% (8 mice per group). DLNs were sectioned to count metastatic cells and were immunostained for CK18+. The metastatic cells from MEER/CD200^High^ tumors were histologically scored compared to metastatic CK18+ cells of MEER/CD200^Low^ tumors, which were assigned a metastatic score of 1. The bar in macroscopic image indicates 100 μm.

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
