# Peer review of "CD200 Induces Epithelial-to-Mesenchymal Transition in Head and Neck Squamous Cell Carcinoma via β-Catenin-Mediated Nuclear Translocation"

_cancers, 2019, doi:10.3390/cancers11101583_

Round 1
Reviewer 1 Report
This manuscript analyzed the role of CD200 in head and neck cancer. It is concluded that CD200 alters EMT-related molecules and increases invasiveness in head and neck cancer through a non-canonical pathway. There are several weak points in the course of the study:
In Fig. 1, the relationship between pathologic differentiation and CD200 was analyzed using TCGA data. However, there is no significance in comparison with the normal group, and the expression of CD200 is increased as grade is increased, but statistical significance compared with the normal group is also excluded. Based on these results, it seems difficult to conclude that the expression of CD200 increases with pathologic differentiation. In addition, in order to conclude that the expression of CD 200 is related to the prognosis of head and neck cancer, the relationship between clinical features and CD 200 expression in TCGA data should be further analyzed. The expression rate of CD200 was low in several head and neck cancer cell lines, but specific experimental evidence is needed. Why did you choose UMSCC84 among many head and neck cancer cell lines? UMSCC84 is a HPV (-) cell line and has a limitation to be compared with a cell line having the same characteristics as the MEER cell line used in animal experiments. Furthermore, verification is required for multiple cell lines, not just one cell line. In Figure 2, the TCGA data should specify what the CD200 + is based on. In Figure. 3, changes in EMT molecules should also be identified along with the phenotype for reduction of invasiveness after treatment with γ-secretase inhibitors.Author Response
Please see the attachment.

Reviewer 2 Report
The manuscript by Shin et al., entitled “CD200 induces epithelial-to-mesenchymal transition in head and neck squamous cell carcinoma via β-catenin-mediated nuclear translocation” examines the role of CD200, a glycoprotein implicated in head and neck SCC (HNSCC), in promoting oncogenic characteristics of HNSCC cells in vitro and in vivo. The significance of CD200 in advanced HNSCC is supported by its enrichment in high grade HNSCC tumors in TCGA, and by previous studies that aligned its expression in basal cell carcinoma with tumor initiation. For the most part, the experimental design is based on functional perturbation of CD200 in one human HNSCC HPV-negative cell line and one murine HNSCC HPV-positive cell line followed by evaluation of key EMT markers and cellular processes associated with stemness and invasiveness in vitro and tumor formation in vivo. Overall, the manuscript is clearly written, interesting and includes novel data suggesting interactions between beta-catenin and the cleaved cytoplasmic domain of CD200. There are a few concerns regarding the experimental design, which introduces confounding factors that may impact data interpretation, such as limited number of cell lines with different HPV status and non-HNSCC tumor microenvironment in mouse studies. In addition, there are some conceptual issues that should be addressed prior to publication.
Major concerns:
1. More cell lines should be used to validate the broader significance of CD200 in head and neck squamous cell carcinoma (HNSCC). The rationale for selecting only one HPV-negative HNSCC cell line, UM-SCC84, for in vitro studies is unclear, since the in vivo studies are carried out with HPV-positive mouse tonsil carcinoma MEER cells, and it is likely to CD200 affects head and neck cancer in via distinct mechanisms in HPV-negative and HPV-positive settings. Thus, it would be useful to examine the effects of altered CD200 expression in a panel of three HPV-negative and three HPV-positive human HNSCC cell lines to gain insights into how HPV impacts CD200 oncogenic activities. Since the majority of head and neck cancer TCGA specimens are HPV-negative, investigation of CD200 in at least three HPV-negative cell lines would generate more convincing results.
2. It is unclear if cell lines were used at defined passages and whether they were routinely authenticated. This information should be included in the Methods section, as cancer cell lines undergo genetic and transcriptional evolution that impacts their responses to inhibitors (Ben-David, Nature, 2018).
3. The idea that the cleaved cytoplasmic tail of CD200 interacts with beta-catenin and impacts its nuclear localization and subsequent transcriptional co-activator activity is intriguing. Does this interaction interfere with the binding of other partners of beta-catenin to its C-terminal domain? Some discussion regarding this issue should be included.
4. Figure 4F showing the cytoplasmic domain of CD200 in complex with beta-catenin in the nucleus erroneously indicates that CD200cyto binds to TCF/LEF at EMT genes’ promoters. This should be corrected.
5. Subcutaneous injection of MEER/CD200High cells does not reflect the tumor microenvironment of head and neck cancer. Would it be possible to carry out these studies using orthotopic injections?
6. The manuscript relies heavily on the premise that EMT is always a feature of cancer stem-like cells and that it is critical for aggressive behavior. Yet, forced expression of CD200 in HNSCC cells only indicates that these cells are capable to acquire more mesenchymal/invasive characteristics when “pushed” to do so in vitro. In the absence of analyses of CD200 enrichment in subpopulations of human HNSCC tumor cells that localize to the invasive tumor front, this assertion may be premature. The Discussion section should include some comments regarding the work of Puram et al. (Cell, 2017) that highlighted partial EMT (p-EMT) as an independent predictor of metastasis and tumor grade in HNSCC.
7. While the experimental strategy employs biased approaches, it would be valuable to assess global transcriptional changes in response to the perturbation of CD200 expression.
Minor concern
For the most part, the results are clearly presented and supported by supplemental figures; the manuscript would be improved if the immunoblot data in Fig. 4 A & D be improved for more convincing results.
Reviewer 3 Report
This manuscript is wellwritten. The autors should add one data and more discuss.
The authors should add the images of beta-catenin-positive nuclei induced by CD200 in Figure 4. The authors should more discuss about other factors that CD200 induces EMT. Beacuse the EMT induced by CD200 is very strong.Author Response
Please see the attachment.

Round 2
Reviewer 1 Report
I agree to accept for this paper.